# Fatty Acids in Waste Tissues: The Nutraceutical Value of Gonads and Livers from the Moroccan *Hypophthalmichthys molitrix* and *Cyprinus carpio* Fishes

**DOI:** 10.3390/md21030188

**Published:** 2023-03-17

**Authors:** Giuseppina Tommonaro, Debora Paris, Giulia Guerriero, Fatima-Zahra Majdoubi, Gaetano Grieco, Carmine Iodice, Lucio Caso, Anouar Ouizgane, Aziz El Moujtahid, Sara El Ghizi, Meriem Bousseba, Mustapha Hasnaoui, Annalaura Iodice, Annabella Tramice

**Affiliations:** 1National Research Council-Institute of Biomolecular Chemistry CNR-ICB, 80078 Pozzuoli, Italy; gtommonaro@icb.cnr.it (G.T.); dparis@icb.cnr.it (D.P.); ciodice@icb.cnr.it (C.I.); lucio.caso@icb.cnr.it (L.C.); annalaura94@live.com (A.I.); 2Comparative Endocrinology Laboratories (EClab), Department of Biology, University of Naples Federico II, 80126 Naples, Italy; grieco.eclab@gmail.com; 3Environmental, Ecological and Agro-Industrial Engineering Laboratory (LGEEAI), Department of Biology, Faculty of Sciences and Techniques, University of Sultan Moulay Slimane, Beni Mellal 23000, Morocco; f.zmajdoubi@gmail.com (F.-Z.M.); az.elmoujtahid@gmail.com (A.E.M.); sara.elg2014@gmail.com (S.E.G.); meriembousseba@gmail.com (M.B.); m.hasnaoui@usms.ma (M.H.); 4Deraoua Fisheries Farm, National Hydrobiology and Fisheries Center, National Agency for Water and Forests, Rabat-Chellah 10002, Morocco

**Keywords:** *Hypophthalmichthys molitrix*, *Cyprinus carpio*, gonadosomatic index (GSI), fatty acids profile (FA), hypocholesterolemic/hypercholesterolemic ratio (h/H), atherogenicity index (AI), thrombogenicity index (TI)

## Abstract

Fishes are an important component of human nutrition, mainly acting as source of essential fatty acids in the prevention of cardiovascular disorders. The increase in their consumption has led to a growth of fishes waste; therefore, the disposal and recycling of waste has become a key issue to address, in accordance with circular economy principles. The Moroccan *Hypophthalmichthys molitrix* and *Cyprinus carpio* fishes, living in freshwater and marine environments, were collected at mature and immature stages. The fatty acid (FA) profiles of liver and ovary tissues were investigated by GC–MS and compared with edible fillet tissues. The gonadosomatic index, the hypocholesterolemic/hypercholesterolemic ratio, and the atherogenicity and thrombogenicity indexes were measured. Polyunsaturated fatty acids were found to be abundant in the mature ovary and fillet of both species, with a polyunsaturated fatty acids/saturated fatty acids ratio ranging from 0.40 to 1.06 and a monounsaturated fatty acids/polyunsaturated fatty acids ratio between 0.64 and 1.84. Saturated fatty acids were found to be highly abundant in the liver and gonads of both species (range 30–54%), as well as monounsaturated fatty acids (range 35–58%). The results suggested that the exploitation of fish wastes, such as the liver and ovary, may represent a sustainable strategy for the achievement of high value-added molecules with nutraceutical potential.

## 1. Introduction

Fish represents an important food source for the countries bordering the Mediterranean, providing bioactive molecules such as fats, proteins, vitamins and minerals which are essential for human health [1,2]. A diet rich in fish promotes cardiovascular health, improves eyesight, prevents arthritis, diabetes, and cancer, protects the brain from diseases, and helps in weight loss [3]. Due to the key role of fish in protecting human health and ensuring a balanced diet, fisheries and aquaculture activities are being increasingly recognized for their essential contribution to global food security and nutrition in the twenty-first century. Aquatic foods remain some of the most traded food commodities in the world, with 225 states and territories reporting some trading activity of fisheries and aquaculture products [4].

Although aquaculture is a limited activity in Morocco—with a low production not exceeding 0.1% of the national fish production—and in the Mediterranean area in general, recently, Morocco and the EU launched a project to strengthen the Moroccan aquaculture sector. In accordance with the objectives of the National Agency for the Development of Aquaculture (ANDA), the goal of this cooperation is centered on the diversification of the fish supply of Morocco and on increasing the fish yield of the country by about 199,000 tons per year [5]. The nine species currently cultured in Morocco belong to six major families of cold water and hot water: *Cyprinidae*, *Salmonidae*, *Esocidae*, *Centrarchidae*, *Cichlids* and *Anguilidae* that are either endemic or non-native to fresh or brackish waters [6]. Carp species (cyprinids) account for about 45% of the total world aquaculture production and are one of the most important fish on the world market, with habitats in rivers (basins) and also in areas largely occupied by marine flora and fauna such as seas, bays, and gulfs where rivers flow [7,8,9]. Compared with other large aquaculture species, such as salmon and shrimp, carp are recognized as an environmentally friendly fish because most are omnivorous (eating mollusks, crustaceans, insect larvae, and seeds) and therefore consume much less fishmeal and fish oil. This makes extensive cultivation highly favored, often in particularly difficult habitats or mixed marine and freshwater areas (such as the Caspian sea or the delta or estuary of rivers) [8,9,10,11,12].

*Cyprinus carpio* (common carp) is one of the most important cyprinid species; it is cultured in over 100 countries worldwide and accounts for up to 10% (over 3 million metric tons) of global annual freshwater aquaculture production [13,14]. This is related to its fast growth rate, easy cultivation in different geographic areas (ponds, canals, rivers, coasts), and its long-life, as well as high feed efficiency and high nutritive value [12,14,15]. Analogously, among the cultivated fishes, *Hypophthalmichthys molitrix* (silver carp) has attracted great attention for its growing production in many Asian countries such as China, Bangladesh, India, Russian Federation, and even in Iran where silver carp is a favorite meal for many people. Silver carp is no less important in the European Union where this species is widespread—produced in particular in Hungary, Romania, Croatia, and the Czech Republic [11,16]. Carp tissues are rich in minerals and vitamins, particularly phosphorous and vitamin B12, as well as having high levels of fatty acids, protein, and antioxidants. This unique nutrient profile makes carp highly desirable for people looking for a better dietary protein; it is also praised as one of the healthiest fish for human consumption [17]. In addition, aiming to promote an eco-friendly aquaculture with a focus on the growing amount of fish waste worldwide, special attention has recently been paid to reusing all fish tissue away from the food market since this represents a rich source of value-added compounds. It is not only fish tissue discarded after capture or fish lost between landing and transport to markets, but also fish tissue not used by the food industry [18].

In a previous study, it was described that waste viscera of common carp were utilized as raw material for the extraction of refined oils by ensilage and fishmeal processes; this carp oil produced a rich source of essential fatty acids of the ω-3 and ω-6 series [19]. Furthermore, not only silver carp meat but also eggs represent a valuable source of essential fatty acids for human nutrition [20]. However, the main therapeutic potential of fish consumption has been ascribed to the presence of polyunsaturated fatty acids (PUFAs) at high concentrations [21]. From a nutritional point of view, it is an established fact that farmed or freshwater fish tissues are a precious source of health-beneficial PUFAs, mainly eicosapentaenoic acid (EPA; C20:5 ω-3), docosahexaenoic acid (DHA; C22:6 ω-3), linoleic acid (LA; C18:2 ω-6), and arachidonic acid (AA; C20:4 ω-6). These ω-3 and ω-6 FAs have a pivotal role in the human diet in preventing many diseases [22,23,24]; in this sense, 2–3 servings of fish per week or the consumption of 250 mg of EPA and DHA per day are recommended by nutritionists [25]. The biological role of fish lipids and, in particular, of their fatty acids sets has been extensively described: lipids are important for fish development because they represent a concentrated source of energy and are involved in many physiological functions (structural component for cellular membrane, precursor of vitamins, hormones, eicosanoids, thermal processes such as osmoregulation and immune response, sexual development) [24,25,26,27,28]. In addition, during ecological investigation, fatty acids—and in particular some polyunsaturated fatty acids—are considered very important biomolecules as trophic biomarkers in the marine food chain analyses of several ecological niches [23]. It is well known that the chemical nature and the percentage of fatty acids in fish tissues vary mainly with fish feeding, but other factors such as size or age, reproductive status, geographic location, season, and temperature influence the fat content and composition of fish muscle [14,24,29]. In a study carried out on gonadal tissues of *Cyprinus carpio*, it was reported that the lipids profile and their concentrations were significantly influenced by salinity and their amount increased with the development of maturity stages [30].

The principal aim of this study was to evaluate the total lipid content—focusing our attention on the analysis of the fatty acid sets—of liver and ovary tissues from immature and mature aquaculture Moroccan fishes of the species *Hypophthalmichthys molitrix* and *Cyprinus carpio* grown in the same conditions before their potential nutritional value was compared with that of their edible fillets [5]. Without making a distinction between material considered waste by the food industry and the edible fish portions, the lipid profiles were compared in order to describe the differences between the two species and the variations related to the stage of sexual maturity and to furnish important information from a nutritional point of view about the fatty acids content of tissues from specific parts of the fish body. Further, tissues were characterized by the hypocholesterolemic/hypercholesterolemic ratio (h/H), the index of atherogenicity (AI), and the index of thrombogenicity (TI). The nutraceutical value of these fish tissues was evaluated, taking into account in particular the possibility of using their gonads and livers (usually not edible parts) for farm food or pharmaceutical applications.

## 2. Results

### 2.1. Fatty Acids Profiles of Common and Silver Carp

The total lipids content and the fatty acid (FA) profiles of the liver, gonads, and fillet tissues of common and silver carp in two different sexual maturation stages were determined, and the results are reported in Table 1. An analysis of total lipid content in all tissues of both carp species showed that at the mature stage, gonads and liver tissues were richer in lipids than those at the immature stage; for mature common carp, the lipid content for 100 g of each tissue was 2.56 ± 0.48 g in the gonads and 1.27 ± 0.22 g in the liver; the corresponding immature tissue amounts were 0.88 ± 0.68 and 0.83 ± 0.08 g, respectively (Table 1). Similar results were also recorded for the mature and immature tissues of silver carp (Table 1). Fillet tissues showed the lowest amount of total lipid in both maturation stages, even though from the immature to the mature stage, the total lipid amount doubled (Table 1). In general, the GC-MS analyses of the FA content from common and silver carp tissues furnished clear information about the recovered fatty acids’ distribution.

#### 2.1.1. Saturated Fatty Acids Profile of Common and Silver Carp

Saturated fatty acids (SFAs) were highly abundant in all tissues, with a preferential accumulation into the liver (~44% in mature and immature common carp; 38% in mature carp; 54% in immature silver carp). However, at the mature stage, the liver content of SFA was almost similar for both species. In contrast, at the mature stage, the SFA amount represented a statistically discriminant parameter for the fillets of the two carps, as reported in Table 1; the SFA content in the fillet samples was 21.37 ± 4.45 mg × 100 g^−1^ of tissues for the mature common carp and a higher value for the corresponding silver carp fillet (65.03 ± 5.91 mg × 100 g^−1^ of tissues, *p* = 0.01).

Moreover, at the immature stage, common carp showed a higher amount of SFA for the liver (*p* = 0.007) and fillet (*p* = 0.03) tissues.

The gonads of mature common and silver carps contained a high amount of SFA; in fact, 207.63 ± 16.61 (36% of total FA) and 170.30 ± 8.52 mg × 100 g^−1^ (31%) of tissues were recovered from the corresponding common and silver carps’ tissues (Table 1).

Palmitic (C16:0) and stearic (C18:0) acids were stored preferentially into the mature gonad and immature liver of both species, as reported in Appendix A.

#### 2.1.2. Monounsaturated Fatty Acids Profile of Common and Silver Carp

Monounsaturated fatty acids (MUFAs) were the most abundant fatty acids: they were recovered preferentially into the livers of the mature (88.41 ± 23.37 mg × 100 g^−1^ of tissues, 48%) and immature stages (209.72 ± 10.01 mg × 100 g^−1^ of tissues, 51%) of common carp and similarly for the silver carp liver (124.79 ± 19.89 mg × 100 g^−1^ of mature tissues, 58%; 52.62 ± 22.00 mg × 100 g^−1^ of immature tissues, 38%).

MUFAs in the gonads were 193.25 ± 18.80 mg × 100 g^−1^ (33%) of mature tissues in the common carp and 252.41 ± 19.79 mg × 100 g^−1^ (46%) of the mature tissues in the silver carp; the corresponding values for immature tissues were lower (Table 1). Firstly, the oleic acid (C18:1ω-9) and then the vaccenic acid (C18:1ω-7) were the main recovered MUFAs.

Evaluating the edible tissues of the fillets, a preference for silver carp was recorded: silver carp mature fillet tissues contained the highest and most significant amount of MUFAs (119.43 ± 3.67 mg × 100 g^−1^ of tissues, 57%, *p* = 0.0006).

For the common carp, the mature gonads and immature liver values of oleic acid were higher than the corresponding tissues of the silver carp, as reported in Appendix A.

For oleic and vaccenic acids, the mature gonad tissues of the common carp provided preferential results. In contrast, only the fillet tissue of the mature silver carp contained 85.47 ± 1.55 mg × 100 g^−1^ of tissues of oleic acid. (Appendix A).

The MUFA/SFA ratio was also calculated for all tissues of the common and silver carp; as reported in Table 1, the values ranged from 0.64 ± 0.33 of silver carp immature fillet to 2.18 ± 0.75 of silver carp immature gonads. However, the gonad tissues of both carps showed values ~1 or above 1 regardless of the growth stage. For the fillet tissues of this common carp, this ratio was always above 1, whereas in silver carp it occurred only at the mature stage. It was interesting to notice that at the mature stage, the gonads of the silver carp exhibited higher and more significant values (*p* = 0.03) than the corresponding common carp, as reported in Table 1.

#### 2.1.3. Polyunsaturated Fatty Acids Profile of Common and Silver Carp

Polyunsaturated fatty acids (PUFAs) preferentially accumulated into the gonad tissues with an impressive difference between the two carp species: 180.36 ± 26.92 mg × 100 g^−1^ of tissues for the mature common carp and 130.93 ± 12.98 mg × 100 g^−1^ for the mature silver carp. In fact, 23–31% of PUFAs was present in the mature gonads of common and silver carp.

In total, 12–34% of PUFAs was recovered in the mature fillets of common and silver carp. Interestingly, the amount of PUFAs was higher in the mature tissues, with 22.75 ± 5.35 mg × 100 g^−1^ of tissues in common carp and 25.73 ± 3.33 mg × 100 g^−1^ of tissues in silver carp.

Liver tissues contained the lowest percentage of PUFAs (4–8%) for both carp species and between the two maturation stages.

On the basis of the proportions of the different fatty acids groups, the PUFA/SFA ratio (Table 1) varied from 0.08 to 1.44. The PUFA/SFA ratio was higher in the gonads and fillets of both carp species. In fact, in the common carp, the mature gonads and fillet tissues values were 0.87 ± 0.15 and 1.06 ± 0.33, respectively, and in the silver carp, the mature gonads and fillet tissues presented corresponding values of 0.77 ± 0.08 and 0.40 ± 0.06.

In Table 1, the ω-3 FA and ω-6 FA values for each tissue are reported. The major polyunsaturated fatty acids identified in Moroccan common and silver carp were C22:6 ω-3 (docosahexanoic acid, DHA, 4.13–14.37%), C20:5 ω-3 (eicosapentaenoic acid, EPA, 3.3–6.25%), C20:4 ω-6 (arachidonic acid, AA, 1.11–13.11%), and C18:2 ω-6 (linoleic acid, LA, 4.40–20.11%).

ω-3 FAs were significantly abundant in the mature gonads of common carp: 119.84 ± 23.40 mg × 100 g^−1^ of tissues (*p* = 0.006); in silver carp gonads, the value was 13.47 ± 0.67 mg × 100 g^−1^ of mature tissues. The mature fillets of common and silver carp contained similar amounts (Table 1), while the corresponding immature tissues were very low in ω-3 FAs.

In contrast, ω-6 FAs were significantly abundant in the mature gonads of silver carp with a value per 100 g of tissues of 117.48 ± 12.31 mg (*p* = 0.04); in common carp gonads, the values were 60.52 ± 3.52 mg of mature tissues and 23.30 ± 1.34 mg of immature tissues. Fillets of common carp contained ~10 mg in 100 g of mature and immature tissues; fillets of mature silver carp contained a value which was higher than the immature fillet amount. 

Arachidonic (AA, C20:4 ω-6) and linoleic (LA, C18:2 ω-6, ω-9) acids were the principal ω-6 FAs detected. In particular, arachidonic acid was preferentially stored into the gonads of common carp, while in silver carp the gonads values were lower and only the mature fillet showed a value comparable to the common carp amount (Appendix A). Linoleic acid was accumulated preferentially in the mature gonad tissues of silver carp.

### 2.2. Fat Quality Indices of Common and Silver Carp

The fat quality of all analyzed tissues was described by means of the following indexes: the hypocholesterolemic/hypercholesterolemic ratio (h/H), the index of atherogenicity (AI), and the index of thrombogenicity (TI). The results are reported in Figure 1, Figure 2 and Figure 3. The h/H ratio ranged from 0.74 to 3.61 (Figure 1); in particular, the h/H ratio of the immature liver tissues of common carp were significantly higher (*p* = 0.005) than the corresponding value in silver carp.

The AI index ranged from 0.23 to 0.53 (Figure 2) and the TI values varied from 0.33 to 1.55 (Figure 3). It was interesting that at the mature stage, the fillet tissues of common and silver carp showed an h/H ratio of 3.44 ± 0.95 and 2.53 ± 0.26, respectively; the common carp mature and immature gonad values were 2.28 ± 0.33 and 3.26 ± 0.25. The AI values of mature common and silver carp fillets were 0.28 ± 0.05 and 0.33 ± 0.03, respectively, and the corresponding TI values were 0.37 ± 0.11 and 0.63 ± 0.07. It is noteworthy that the AI values of the immature common and silver carp gonads were equally low and equal to ~0.3. The AI (*p* = 0.01) and TI (*p* = 0.03) indexes of the mature fillet of common carp were significantly low. The TI values were higher than 1 only for the liver tissues of both species.

## 3. Discussion

It is well established that the consumption of fish or fish products containing bioactive lipids (MUFA, PUFA, lipid-soluble vitamins A, E, and D, natural antioxidants) has several health benefits, including a reduced risk of cardiovascular disease (CVD) and coronary heart diseases and prevention of cardiac arrhythmias, as well as anti-inflammatory and anti-thrombotic potency, and also efficient action to counter a plethora of diseases characterized by chronic inflammation (e.g., cancer) [31,32,33].

Fat quality in immature and mature Moroccan aquaculture *Hypophthalmichthys molitrix* and *Cyprinus carpio* tissues was described in this paper. Particular attention was devoted to those tissues normally not utilized in the production of human food such as the liver and gonads, but that could be used as feed for the production of super-fish or as a source for nutraceutical production, or even for industrial food preparation due to their relevant molecular profile. These fishes were collected from the Deroua Fisheries Station located in the Beni Mellal-Khenifra region, grown in the same earthen ponds, and equally naturally fed. The gonads and liver tissues of carp samples from mature (common carp GSI 18.34 ± 1.12; silver carp GSI 0.67 ± 0.26) and immature fishes (common carp GSI 7.08 ± 1.94; silver carp GSI 0.08 ± 0.03) (Table 2) were selected and investigated for their total lipid content and compared with fillets’ lipid profiles.

The fatty acids profiles were calculated and examined, particularly their content of saturated fatty acids (SFAs), monounsaturated fatty acids (MUFAs), and polyunsaturated fatty acids (PUFAs) with attention to the amount of ω-3 and ω-6 fatty acids, the ratios PUFA/SFA and MUFA/SFA, the hypocholesterolemic/hypercholesterolemic ratio (h/H), and the indexes of atherogenicity (AI) and of thrombogenicity (TI).

The important result was that those tissues considered as waste or normally not edible (gonads and livers) proved to be an important source of fatty acids and, properly dissecting carp viscera, it was possible to obtain extracts with different fatty acids profiles. It was interesting to note that in the passage from the immature to the mature (adult) stage, an increase in the lipidic content was observed for all tissues, but preferentially in the gonads and liver tissues of both carp species, the accumulation that resulted was remarkable. As reported in Table 1, the liver and the gonads of mature common carp contained, respectively, 2.56 and 1.27 g × 100 g^−1^ of tissues of total lipids, whereas in the immature stage, the corresponding values were 0.88 and 0.83 × 100 g^−1^ of tissues. We observed a mobilization of energy molecules of Moroccan common and silver carp during sexual maturation toward the liver and the gonads tissues, presumably in support of reproductive efforts. These results were in agreement with those previously reported; in fact, the liver constitutes the location for fat deposits in all fish species [34] and, during the maturation stage, a transfer of fats and also proteins to the reproductive sites has been extensively documented [35]. Furthermore, lipids are the main source of metabolic energy in marine fish for swimming, growth, and reproduction, and lipid levels vary in relation to the reproductive cycle [14,35,36].

On the other hand, as described before, fish lipid content and fatty acid (FA) profiles are greatly dependent on the growth conditions such as the feeding conditions, the seasonal variation, the geographical area of recovery, the reproductive status, and the farmed or wild origins [14,37,38,39]. Herein, the environmental (temperature, salinity, pH, geographical area) and feeding conditions were the same for all samples. At Deroua fish farm (Beni-Mellal, Morocco), phytoplankton constituted the basic diet of silver carp in ponds. The detritivorous carp (common carp) consumed essentially the organic matter contained in pond sediments. Therefore, the difference in fatty acids profiles depended on the reproductive status and on the distinct species.

Analyses of the total FA amounts recovered from the three selected tissues revealed some differences between the common and silver carp.

At the mature stage, gonads and liver tissues showed almost the same content of total FAs for both species, the total FA content of silver carp fillet tissues was 0.210 ± 0.008 g × 100 g^−1^, which was higher than the value of 0.067 ± 0.008 g × 100 g^−1^ of the corresponding common carp fillets.

Interestingly, the composition of the FA mixtures depended on the tissues and maturation stage of the common and silver carp. In general, saturated fatty acids (SFAs) were recovered at 24–54% of the investigated carp tissues. In contrast, the content of total MUFAs ranged from 33 to 58% of the total FA; they were the most abundant fatty acids and the values here shown are higher than those previously reported [40].

The MUFAs content in mature silver carp was higher than in common carp for all tissues and in particular from the statistical analyses, the MUFA values of mature liver and fillet tissues of silver carp were favored.

These results appeared relevant especially when compared with data previously shown. Jorjani et al. (2015) [38] described the fatty acid profile of fillets from Iranian cultured common carp and silver carp, which were cultured in a semi-intensive manner with natural feeding: the MUFA contents of common and silver carp were evaluated to be 45% and 33% of the total FAs, respectively. In our analyses, the silver carp mature fillet tissues contained 19.43 ± 3.67 mg × 100 g^−1^ of tissues, which corresponded to 57 % of the total FAs [41].

The reason for the difference between our results regarding both SFA and MUFA values and the literature data is likely due to the different eating habits of the Moroccan fishes [14,41].

The MUFA/SFA ratio for the mature tissues of common and silver carp was generally above 1; this was interesting because this index was assessed in monitoring the fish growth in different seasonal conditions [14,42]. However, this ratio was a statistically discriminating parameter for both species: in fact, at the mature stage, the silver carp gonads showed higher values.

Aiming to propose a sustainable consumption of adult fishes’ meat, common carp meat resulted in generally less fat, but the percentage of MUFAs in the mature fillets of silver carp was higher and with almost the same amount of PUFAs for both species. In fact, the PUFAs of Moroccan common and silver carps ranged from 3.53 to 36.82% of the total FAs, with 12–34% of PUFAs which were recovered in the mature fillets of common and silver carp. Evaluating the fillet tissues of both species, at the mature stage, the amount of total ω-3 FA was similar with a value of ~13 mg × 100 g^−1^ of tissues; analogous to the total ω-6 FA with a value of ~11 mg in 100 g of tissues. These results are of great relevance from a nutritional point of view because fillets represent the preferential edible part of fishes. The PUFAs content of fillets (values from 12.24 to 34.22%) and the PUFA/SFA ratio (0.40–1.06, Table 1) were comparable to values recovered from farmed carp species which were grown in different conditions [14,19,39]. These values are in agreement to those recommended by the World Health Organization (WHO) and the Food and Agriculture Organization (FAO) [39,43]. In fact, according to the FAO, WHO, the British Department of Health (1994), and other authors and literature sources, PUFA/SFA ratios should preferably be within the range of 0.35 to 1.0 [44,45,46,47]. However, these values were around 1, or greater than 1 in the case of gonad tissues. Yeganeh et al. (2012) [14] reported that one of the marked differences between the farmed and wild common carp is the higher level of linoleic acid present in the farmed fish (about 15.3 vs. 3.1%). In the current analyses, linoleic acid was recovered in fillet tissues for ~3% for both species, similar to the wild species. In fact, this compound was present in the plant oils which are used in the feeding of farmed fish and accumulate largely unchanged in the lipids of marine fish [48]. In agreement with data previously reported for wild carp collected in different seasons [14,42], the fillets of both Moroccan species showed a good amount of arachidonic acid (AA, ~5 mg × 100 g^−1^ of mature tissues). The h/H ratio and AI and TI indices were calculated for the first time not only for fillets but also for the gonads and liver of both species, as shown in Figure 1, Figure 2 and Figure 3. The results on lipid indices reported in this study are in agreement with those previously reported on the carp species [24,47,49,50].

The AI index describes the ability of pro-atherogenic activity or preventive anti-atherogenic effect (inhibiting the aggregation of plaques, diminishing the levels of cholesterol and phospholipids, and thus preventing coronary diseases). TI expresses the tendency to form clots in the blood vessels [50,51]. In all cases for the tissues of common and silver carp, AI was estimated to be below the recommended value of 1.0 [52] (Figure 2), with an interestingly low index of 0.28 and 0.33 for the common and silver carp fillets, respectively. Moreover, the h/H index was always above 1 for the mature and immature gonads and fillet of both species, and it ranged between 1.84 and 3.61. The TI values were higher than 1 only for the liver tissues of both species. These values below 1.0 provided evidence of good antithrombogenic properties. The TI values were determined from three saturated fatty acids: myristic (C14:0), palmitic (C16:0), and stearic (C18:0), although palmitic and stearic acid were the most abundant in our analyses.

In general, the low AI and TI values, as well as the high h/H index recorded for the silver and common carp tissues, had to be considered as important results because they are recommended in a healthy diet for the prevention of cardiovascular disorders [46,47,53]. It was worth noting that significantly lower AI and TI values were observed in in the fillet of mature common carp.

### 3.1. Fatty Acid Profile of Gonads of Common and Silver Carps

The fatty acids analysis of mature gonads tissues from common and silver carp revealed that they represent an important stock site of the total saturated fatty acids (SFAs). Table 1 shows that the highest value of SFAs was determined for the mature gonads of the common carp (207.63 ± 16.61 mg × 100 g^−1^ of tissues), while 170.30 ± 8.52 mg × 100 g^−1^ was the corresponding value of the silver carp mature gonads. The corresponding values at the immature stage were 13–15% lower.

As it has been described that MUFAs are involved in gonadal development [54], here we reported that the accumulation of MUFAs was greater in the mature gonads tissues of common and silver carp (193.25 ± 18.80 and 251.41 ± 19.79 mg × 100 g^−1^ of tissues, respectively), as a parameter of good fish living conditions. Furthermore, although the MUFA values of the mature gonads of the two carp species were high, they showed a significant difference favoring the silver carp tissues. In particular, while a prevalence of oleic and vaccenic acids was recorded in all tissues of both species, the gonad tissues of mature silver carp contained a noticeable amount of oleic acid. The importance of oleic acid is related to its properties in preventing cardiovascular diseases and its beneficial effect on cancer, autoimmune, and inflammatory diseases, in addition to its ability to facilitate wound healing [55,56]. High levels of oleic, and also arachidonic, acid have been reported as a characteristic property of freshwater fish oils [57].

A preferential and extremely important accumulation of PUFAs in the mature gonad tissues of Moroccan common and silver carps was recorded, although with some differences. A statistical comparison of the ω-3 and ω-6 values in both species, showed that ω-3 FAs were significantly abundant in the mature gonads of common carp, whereas ω-6 FAs were prevalent in the mature gonads of silver carp. The production of fish oils from carp viscera containing fatty acids of the ω-3 and ω-6 series has been previously reported [19]; further to this, the selection of the visceral tissues ensures the possibility of preparing oils with different ω-3 and ω-6 compositions.

The presence of PUFAs in the gonads of both mature species, and also an understanding of where and which FAs are allocated within the carp body (in particular the ω-3 and ω-6 FAs), could help to design efficient aquaculture systems and particularly to increase the ω-3 FAs in edible tissues and/or advise on how to reuse fish tissues for subsequent feeds rich in ω-3 FAs. In fact, it has been reported that more EPA, DHA, and in general PUFAs are present in fish fed with fish oil supplements compared with fish fed with vegetable oils [58]. EPA and DHA are needed for normal growth and development. They also affect retinal and brain phospholipid composition, intelligence quotient (IQ), and motor development [59]. The positive effect of ω-3 PUFAs on coronary heart diseases has been shown in many experimental studies in animals, humans, and tissue culture [60,61].

However, we recorded a preferential accumulation of AA in the gonad tissues of common carp. This is an important result due to the biological role of this compound. AA is a necessary FA because it is the precursor of prostaglandin and thromboxane, which will influence blood clot formation and its attachment to the endothelial tissue during wound healing. Moreover, it plays a key role in growth [62]. Evaluating the h/H ratio and AI and TI indices for gonad tissues, as shown in Figure 1, Figure 2 and Figure 3, it was observed that the index (h/H) was always above 2 and the AI was estimated to be ~0.3 for common carp gonads. The TI index was less than 1 for the mature and immature gonads of both carps (Figure 3). These data furnish clear clues on the possibility of better exploiting these tissues in the food or nutraceutical fields [63].

### 3.2. Fatty Acid Profile of Livers of Common and Silver Carps

The data recorded suggested that during the maturation stage, common and silver carp liver tissues lost a percentage of their fatty acids content (in particular common carp), although the total lipid content (saponificated and non-saponificated lipids) increased significantly In fact, if SFA values at the mature stage were low for both species, at the immature stage their values were remarkable: the second highest value of SFA was determined for the immature liver of common carp (187.80 ± 13.84 mg × 100 g^−1^ of tissues), which represented an important stock site of these compounds and which was statistically different from the corresponding value of silver carp liver (*p* = 0.008). These results were in agreement with data obtained from carp collected in other geographical areas [14]. Analogously, MUFAs decreased their values in the livers of mature common carp; instead, PUFAs values were generally low and similar in both maturation stages. These results suggested a redistribution of the lipid supplies during sexual maturation, meaning that they could be used with the aim to reuse these tissues or to monitor fish development.

## 4. Materials and Methods

### 4.1. Sample Collection and Preparation

*Hypophthalmichthys molitrix* (silver carp) and *Cyprinus carpio* (common carp) were recovered from the Deroua Fisheries Station located in the Beni Mellal-Khenifra region, a semi-arid climate zone in Morocco. The collection was conducted in April 2019. Fishes were both held in earthen ponds. Phytoplankton constituted the basic diet of silver carp in ponds. The detritivorous carp (common carp) consumed essentially the organic matter contained in pond sediments [64,65].

The breeding season in Morocco of the studied species, which is classified in arid and semi-arid zones, starts from April and extends to June when the water temperature in ponds varies between 20 °C and 26 °C. The seining of fish was performed smoothly in order to avoid fish stress. Female fish were recognized by their inflated and soft abdomen, inflated anus, and red pectoral fins. Mature silver carp were selected from 46 cm in length and up [66]; common carp female were selected from 54 cm in length and up [67] (see Mutethya et al., 2020). The mean of the biometrical data is reported in Table 2. The data include total weight, length, body circumference, gonadal weight, liver weight, fillet weight, and the gonadosomatic index (GSI, gonadal weight × 100)/total weight) reported as confirmation of fish maturity [68]. Our experiments were performed in strict accordance with European (Directive 2010/63) legislation on the care and use of animals for scientific purposes.

After being captured by seine nets, the specimens were anaesthetized by 2-phenoxyethanol and transported in an ice box at 4 °C within 3 h, and the biometric data of each dead fish were recorded: total weight, length, diameter, gonadosomatic index. Each sample was carefully subjected to abdominal dissection for the elimination of the viscera and for recovering its gonads, liver, and fillet tissues, and the collected materials were stored at −20 °C until molecular processing. All samples were treated under the same conditions. At the beginning of each lipid profile analysis, the aliquots of 10 samples for each species and maturity stage were permitted to equilibrate to room temperature, ground, and homogenized.

### 4.2. Lipid and Fatty Acid Analysis by Gas Chromatography-Mass Spectrometry (GC-MS)

Lipids were extracted from the carp tissues according to the Bligh and Dyer method [69]. A total of 50–100 mL of a solution of dichloromethane: methanol: water (10:20:7.5 *v*/*v*/*v*) was added to the tissues and lipid release was promoted by crumbling them in a mortar for 5–10 min as previously described [23].

The qualitative and quantitative characterizations of the total fatty acids recovered from the fish tissue organic extracts were determined by GC–MS on the corresponding fatty acid methyl esters (FAMEs); they were obtained after the saponification of lipid extracts and following a modified AOAC Official Method, 991.39 [70], as previously reported [23]. The lipid fraction was saponificated and the free fatty acids were derivatized to fatty acid methyl esters (FAMEs) by methanolic sodium methoxide anhydrous (2–6 mL, 1 N) at 90 °C for 55 min [23,71]. All operations involving lipids or their constituent fatty acids were conducted in an atmosphere of pre-purified nitrogen. FAMEs were recovered by adding diethyl ether to the cooled reaction mixture. The FAME mixtures were redissolved in diethyl ether (EE, 2 mg/mL) and analyzed by GC–MS (scan mode) equipped with an ion-trap (Thermo Scientific, Waltham, MA, USA) on a 5% diphenylpolysiloxane column (OV-5 column, VF-5ms 30 × 0.25, Agilent Technologies, Middelburg, The Netherlands) able to separate FAMES, as reported in Appendix A. A Thermo Scientific™ PolarisQ ™ GC/MSn Benchtop Ion Trap Mass Spectrometer was used in EI (70 eV) under positive mode analysis (mass range 50–450). The elution of free fatty acid methyl esters required an increasing temperature gradient, as previously reported [23]. Samples of 2 μL were directly injected in split mode (1:10) and at a split flow of 10 mL min^−1^, with a blink window of 3 min (with an inlet temperature of 270 °C, the transfer line set at 280 °C, and an ion-source temperature of 250 °C). The carrier gas was helium, which was used at a constant flow of 1.0 mL min^−1^. FAMEs were identified by comparison of their retention time with those present in a mixture of 11 standard FAMEs (Appendix A, marine source, analytical standard, Sigma Aldrich) which was analyzed under the same conditions. 

Aiming to quantitatively characterize each fish fatty acid mixture, for each GC–MS measurement, an internal standard was added to the FAMEs solution to be analyzed; a total of 50 μL of a 2.5 mg mL^−1^ solution of methyl tricosanoate (C23:0) was used, corresponding to 0.125 mg of internal standard per mL of solution to be analyzed (0.454 μg for each injection of 2 μL).

The equation:CFA= [(m_IS_ × A_FA_ × RRF_FA_)/(1.04 × A_IS_)]/V_-injection_
was used to quantify all measured fatty acids, where CFA is the concentration of the fatty acid in mg × mL^−1^, V_-injection_ is the volume of each injected sample solution (2 μL), m_IS_ is the weight of the internal standard, A_FA_ is the fatty acid peak area in the GC spectrum, RRF_FA_ is the relative retention factor for each fatty acid [70], 1.04 is the correlation factor between the fatty acids and fatty acid methyl esters, and A_IS_ is the internal standard peak area in the GC spectrum. The RRF value accounts for the effective carbon number, and it was calculated according to previously published methods [70,71]: myristic acid, C14:0 (RRF = 1.08); palmitic acid, C16:0 (RRF = 1.05); palmitoleic acid, C16:1 n-7 (RRF = 1.05); stearic acid, C18:0 (RRF = 1.04); oleic acid, C18:1 n-9 (RRF = 1.03); eicosapentaenoic acid (EPA), C20:5 n-3 (RRF = 0.99); docosahexaenoic acid (DHA), C22:6 n-3 (RRF = 0.97); tricosanoic acid, C23:0 (RRF = 1.000).

As previously reported [23,70,72,73], the quantitation of individual FAs is based on the comparison of their peak areas, A_i_, and the peak area of a suitable IS (internal standard), A_IS_, which was in our case tricosanoic acid (23:0), the usual standard for the determination of PUFAs in fish extracts.

The response factors for each of the FAMEs not present in the standard mixture were estimated by comparison with the standard FAMEs which resembled them most closely in terms of chain length and number of double bonds. In any case, the validity of the theoretical relative RF (RRF) [24] was confirmed by Craske and Bannon [74] and Erder [75].

Furthermore, starting from CFA, it was possible to establish the total amount of each fatty acid (FA) as either free or saponificated fatty acid which was recovered in the EE extract of each fish species, as previously reported [23,70]; according to the weight of the frozen fillet fish tissue (m_fis_), the mgs of each fatty acid free in 100 g of fish tissue were determined by the following formula:[(mg FA in 1 mg of EE extract × total mg of EE extract)/g m_fis_] × 100
where g m_fis_ corresponds to the grams of frozen fish tissue analyzed. Three replicates of each sample were obtained. A total of 16 fatty acids were measured, with a limit of quantification (LOQ) of 0.3 mg/100 g of fish tissue.

### 4.3. Determination of Fat Quality Indices

The hypocholesterolemic/hypercholesterolemic ratio (h/H), the index of atherogenicity (AI), and the index of thrombogenicity (TI) were determined to describe the fat quality.

These factors were calculated using the following equations [49,50,76]:
**h/H** = Σ(C18:1n 9, C18:1n 7, C18:2n 6, C18:3n 6, C18:3n 3, C20:3n 6,C20:4n 6, C20:5n 3, C22:4n 6, C22:5n 3, C22:6n 3)/Σ(C14:0, C16:0)
**AI** = (C12:0 + 4 × C14:0 + C16:0)/((n − 6)PUFA + (n − 3)PUFA + MUFA)
**TI** = (C14:0 + C16:0 + C18:0)/(0.5 × MUFA + 0.5 × (n − 6)PUFA + 3.0×(n-3)PUFA + (n − 3)PUFA/(n − 6)PUFA).

### 4.4. Statistical Analysis

The results obtained were presented as the means of measurements and the standard error (SE). Differences in the fatty acids values of the gonads, liver, and fillet tissues of fish studied were estimated with a Student’s *t* test: for each lipid species, we assessed the differences between common and silver carp within the same growth stage, i.e., mature or immature with R software R core team (https://www.r-project.org/ accessed on 4 February 2023). *p* values < 0.05 were considered as statistically significant.

## 5. Conclusions

Carp are considered fish with great adaptability because they are able to grow not only in freshwater habitats but also in areas occupied by marine flora and fauna (ponds, rivers, sea coasts, etc.). An analysis of the fatty acid profiles of Moroccan common and silver carps, generally targeting the consumption of fishes at the mature stage, was reported in this study. The data suggested that the PUFA content of the edible parts of both carp species at the mature stage was interesting from a nutritional point of view, with AI and TI values below 1. Although there was a significant presence of biologically relevant FAs (EPA, DHA, AA) in both mature fillet species, the content of SFA in mature fillets and the PUFA/SFA ratio resulted in discriminating parameters for both carps, with the common carp values being preferable and better than those of the silver species.

The most important result of this study was that the fatty acid profiles of non-edible parts of Moroccan common and silver carps were determined: it revealed the possibility of using selected visceral tissues such as livers and gonads for the production of fish oil with a different composition or fishmeal from fisheries by-products. This would be extremely attractive as it would greatly help to reduce industrial fish waste materials, thereby promote environmental protection, economic growth, and human health.

In fact, the majority of worldwide fish oil production is mostly used in the aquaculture industry, while only a small proportion is used for the production of ω-3 PUFA-related products. Thus, the fishing of most species just for the production of fish oil is not a sensible or sustainable approach. Instead, reusing fish residues and side streams of processing, such as the head, liver, ovary, skin, trimes, etc., is considered a sustainable circular economy strategy, since fish by-products contain lipid ingredients and bioactive compounds.

Suffice it to say that PUFAs were present in the mature gonads of common and silver carp for 23–31% of the total FAs; ω-3 FAs were abundant in the mature gonads of common carp, whereas silver carp mature gonads contained ω-6 FAs preferentially.

In summary, our results provided some clues on the possibility of recycling fish body parts not used for food preparations as super-feeds and/or for obtaining fish extracts with high added-value and with different compositions, which can be employed for applications in human health and other industries (i.e., aquaculture, food, agrochemical, biotechnological, and pharmaceutical applications).

## Figures and Tables

**Figure 1 marinedrugs-21-00188-f001:**
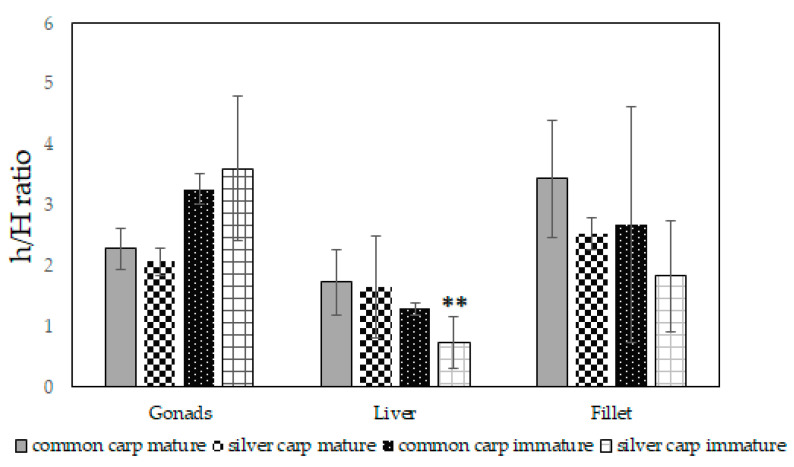
The hypocholesterolemic/hypercholesterolemic ratio (h/H) of immature and mature gonads, liver, and fillet tissues of common carp (*Cyprinus carpio*) and silver carp (*Hypophthalmichthys molitrix*). For each tissue, significant discrepancies were evaluated with a Student’s *t* test comparing the two fish species within the same growth stage. ** *p* < 0.01.

**Figure 2 marinedrugs-21-00188-f002:**
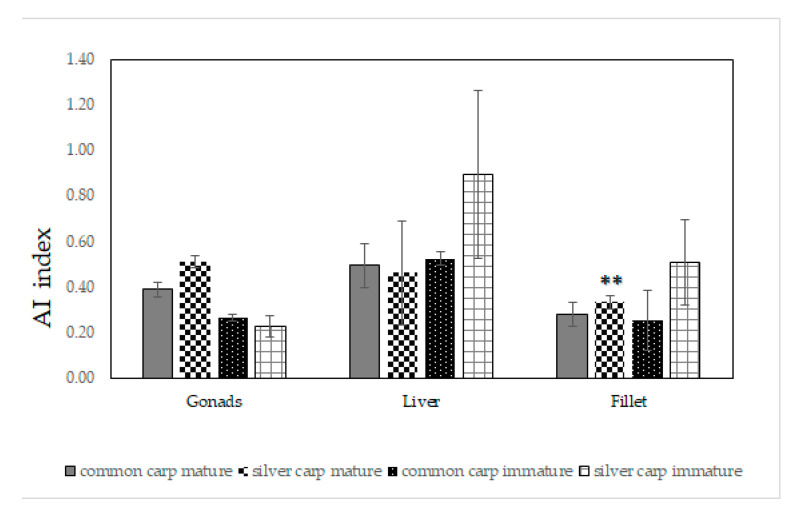
The index of atherogenicity (AI) of immature and mature gonads, liver, and fillet tissues of common carp (*Cyprinus carpio*) and silver carp (*Hypophthalmichthys molitrix*). For each tissue, significant discrepancies were evaluated with a Student’s *t* test comparing the two fish species within the same growth stage. ** *p* < 0.01.

**Figure 3 marinedrugs-21-00188-f003:**
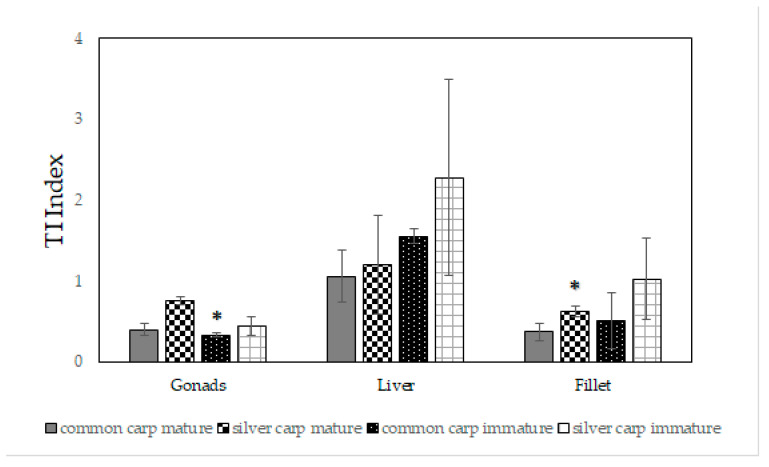
The index of thrombogenicity (TI) of immature and mature gonads, liver, and fillet tissues of common and silver carps. For each tissue, significant discrepancies were evaluated with a Student’s *t* test comparing the two fish species within the same growth stage. *****
*p* < 0.05.

**Table 1 marinedrugs-21-00188-t001:** Total fat, total FA, SFA, MUFA, PUFA, ω-3 FA, and ω-6 FA concentrations (mean mg × 100 g^−1^ of tissue ± SE), PUFA/SFA and MUFA/SFA ratios of the gonads, liver, and fillet tissues of immature and mature common carp (*Cyprinus carpio*) and silver carp (*Hypophthalmichthys molitrix*).

	Total Lipids (g/100 g)	Total FA ^a^ (g/100 g)	SFA ^b^ (mg/100 g)	MUFA ^c^ (mg/100 g)	PUFA ^d^ (mg/100 g)	ω-3 ^e^ FA (mg/100 g)	ω-6 ^f^ FA (mg/100 g)	PUFA/SFA	MUFA/SFA
**common carp immature**									
**G**	0.88 ± 0.68	0.109 ± 0.006	28.10 ± 1.22	41.14 ± 3.45	40.36 ± 2.14	17.06 ± 0.80	23.30 ± 1.34	1.44 ± 0.10	1.46 ± 0.14
**L**	0.83 ± 0.08	0.412 ± 0.006	187.80 ± 13.84	209.72 ± 10.01	14.56 ± 2.31	n.d.	14.56 ± 2.31	0.08 ± 0.01	1.12 ± 0.10
**F**	0.14 ± 0.07	0.059 ± 0.033	17.09 ± 9.05	29.64 ± 19.08	12.81 ± 5.66	4.70 ± 1.54	8.11 ± 4.12	0.75 ± 0.52	1.73 ± 1.44
**silver carp immature**									
**G**	0.25 ± 0.12	0.109 ± 0.033	26.43 ± 5.06	57.72 ± 16.45	25.05 ± 5.58 *****	7.13 ± 0.97 ******	17.92 ± 4.61	0.95 ± 0.28 *	2.18 ± 0.75
**L**	0.44 ± 0.18	0.135 ± 0.067	73.20 ± 27.91 **	52.62 ± 22.00 *	9.33 ± 5.00	n.d	9.33 ± 5.00	0.13 ± 0.08	0.72 ± 0.41
**F**	0.25 ± 0.06	0.021 ± 0.008	10.23 ± 3.61 *	6.50 ± 2.47	4.20 ± 1.65	1.60 ± 0.52 *****	2.60 ± 1.13	0.41 ± 0.22	0.64 ± 0.33
**common carp mature**									
**G**	2.56 ± 0.48	0.581 ± 0.056	207.63 ± 16.61	193.25 ± 18.80	180.36 ± 26.92	119.84 ± 23.40	60.52 ± 3.52	0.87 ± 0.15	0.93 ± 0.12
**L**	1.27 ± 0.22	0.182 ± 0.058	79.79 ± 13.49	88.41 ± 23.37	14.64 ± 3.61	8.89 ± 2.01	5.74 ± 1.60	0.18 ± 0.05	1.11 ± 0.35
**F**	0.28 ± 0.12	0.067 ± 0.014	21.37 ± 4.45	22.38 ± 4.28	22.75 ± 5.35	13.57 ± 3.04	9.19 ± 2.31	1.06 ± 0.33	1.05 ± 0.30
**silver carp mature**									
**G**	0.98 ± 0.27	0.553 ± 0.055	170.30 ± 8.52	252.41 ± 19.79	130.96 ± 12.98 *****	13.47 ± 0.67 ******	117.48 ± 12.31 *	0.77 ± 0.08	1.48 ± 0.1 *****
**L**	1.10 ± 0.47	0.215 ± 0.091	80.72 ± 39.48	124.79 ± 19.89 *	9.45 ± 5.03	n.d	9.45 ± 5.03 *	0.12 ± 0.08	1.55 ± 0.79
**F**	0.50 ± 0.08	0.210 ± 0.008	65.03 ± 5.91 *****	119.43 ± 3.67 ******	25.73 ± 3.33	12.01 ± 1.36	13.72 ± 1.98	0.40 ± 0.06	1.84 ± 0.1

G: gonad tissues; L: liver tissues; F: fillet tissues; ^a^ Total fatty acids recovered from different tissues. ^b^ Total SFA in tissues, including, C14:0 (myristic acid), C16:0 (palmitic acid), C17:0 (margaric acid), C18:0 (stearic acid), C20:0 (arachidic acid), C22:0 (behenic acid).^c^ Total MUFAs in tissues, including, C16:1 ω-7 (palmitoleic acid), C17:1 (margaroleic acid), C18:1 ω-7 (vaccenic acid), C18:1 ω-9 (oleic acid), C20:1 ω-9 (gadoleic acid), and C22:1 (erucic acid). ^d^ Total PUFAs in tissues; the most abundant were evaluated, corresponding to ω-3, ω-6 fatty acids. ^e^ Total ω-3 fatty acids in tissues including C20:5 ω-3 (EPA) and C22:6 ω-3 (DHA); ^f^ Total ω-6 fatty acids, including 18:2 ω-6 (linoleic acid) and 20:4 ω-6 (arachidonic acid). The statistical significance of lipids was assessed in gonads, liver, and fillets via a Student’s *t* test comparing the two fish species within the same growth stage. Abbreviations: *****
*p* < 0.05; ** *p* < 0.01; n.d.: not detected.

**Table 2 marinedrugs-21-00188-t002:** Biometrical measurements and maturity index (GSI, gonadal weight × 100)/(total weight) of ten immature and mature female carps, *Hypophthalmichthys molitrix* (silver carp) and *Cyprinus carpio* (common carp) collected at the Deroua Fisheries Station, Beni Mellal-Khenifra region (Morocco).

Female Fish (*n* = 10 Species)	Total Weight (g)	Length (cm)	Body Circumference (cm)	Gonadal Weight (g)	Liver Weight (g)	Fillet Weight (g)	GSI (%)
Immature *Hypophthalmichthys molitrix* (silver carp)	456.80 ± 70.20	36.01 ± 2.59	16.66 ± 1.13	0.52 ± 0.20	2.54 ± 0.54	19.62 ± 3.29	0.08 ± 0.03
Mature *Hypophthalmichthys molitrix* (silver carp)	1845.23 ± 573.06	53.50 ± 5.18	26.75 ± 3.73	8.23 ± 1.18	13.60 ± 4.87	53.08 ± 16.60	0.67 ± 0.29
Immature *Cyprinus carpio* (common carp)	804.28 ± 61.94	39.75 ± 0.78	24.13 ± 0.83	74.03 ± 6.59	11.58 ± 1.74	26.93 ± 4.65	7.08 ± 1.94
Mature *Cyprinus carpio* (common carp)	2800.07 ± 258.80	52.53 ± 2.48	39.33 ± 0.33	519.37 ± 80.10	12.07 ± 0.55	56.87 ± 8.03	18.34 ± 1.12

## Data Availability

The data presented in this study are available on request from the corresponding authors.

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
