# Peer review of "Fatty Acids in Waste Tissues: The Nutraceutical Value of Gonads and Livers from the Moroccan *Hypophthalmichthys molitrix* and *Cyprinus carpio* Fishes"

_marinedrugs, 2023, doi:10.3390/md21030188_

Round 1
Reviewer 1 Report (Previous Reviewer 1)
The manuscript has been improved but additional revision is required.
The authors stated that EPA, DHA, AA are essential fatty acids. The essential fatty acids for humans are linoleic and linolenic acids. Please see lines 439 and 592
The authors failed to show statistical comparison among results. There is no explanation about discrepancies obtained.
The RRF reported in the manuscript were based on Cladis, D. P., Kleiner, A. C., Freiser, H. H., & Santerre, C. R. (2014). Fatty acid profiles of commercially available finfish fillets in the United States. Lipids, 49(10), 1005-1018 that were obtained by GC/FID, as well as the factors reported by Tvrzická, E., Vecka, M., Staňková, B., & Žák, A. (2002). Analysis of fatty acids in plasma lipoproteins by gas chromatography–flame ionization detection: Quantitative aspects. Analytica Chimica Acta, 465(1-2), 337-350.
The formulas should be reviewed.
Author Response
Q1: The manuscript has been improved but additional revision is required.
A1: Many thanks for your valuable opinion.
Q2: The authors stated that EPA, DHA, AA are essential fatty acids. The essential fatty acids for humans are tha linoleic and linolenic acids. Please see lines 439 and 592.
A1: Thank you for your comment. You are right about the linoleic and linolenic acids and at the lines 439 and 592 the adjective “essential” has been changed in “necessary” and “ biologically relevant”.
Q3: The authors failed to show statistical comparison among results. There is no explanation about discrepancies obtained.
A3: We thank the reviewer for his constructive criticisms. As we just described data comparisons with the general formula for the discrepancy between measures, considering data as means ± standard error, we now revised the statistical approach such to better define our results. We reformulated statistical data analysis, and reported comparisons among classes with relative p values applying Student’s t test to assess significant differences. Modifications are reported in the text and table1.
Q4: The RRF reported in the manuscript were based on Cladis, D. P., Kleiner, A. C., Freiser, H. H., & Santerre, C. R. (2014). Fatty acid profiles of commercially available finfish fillets in the United States. Lipids, 49(10), 1005-1018 that were obtained by GC/FID, as well as the factors reported by Tvrzická, E., Vecka, M., Staňková, B., & Žák, A. (2002). Analysis of fatty acids in plasma lipoproteins by gas chromatography–flame ionization detection: Quantitative aspects. Analytica Chimica Acta, 465(1-2), 337-350. The formulas should be reviewed.
A4: Thank you for your valuable suggestion. The formula has been revised. See the text .

Reviewer 2 Report (Previous Reviewer 2)
The manuscript has been well improved.
Author Response
Dear reviewer we would like to thank you for your decision.
Reviewer 3 Report (Previous Reviewer 3)
The MS was corrected. I found only few minor editorial mistakes which I marked in the text. I recommend accept the MS after minor revision.

Author Response
Reviewer3:
The MS was corrected. I found only few minor editorial mistakes which I marked in the text. I recommend accept the MS after minor revision.
A1: Thanks for your comment. The text has been correct.
Reviewer 4 Report (Previous Reviewer 4)
Authors have addressed my earlier comments.
Author Response
dear reviewer we would like to thank you for your decision.
Round 2
Reviewer 1 Report (Previous Reviewer 1)
Accept
Author Response
Here are the Academic Editor's comments:
1. The result description part and the experimental method part of this paper are too complicated and should be simplified as much as possible.
Thank you for your opinion. Result description and materials and methods have been shortened.
- The core content of this article is the analysis of lipids, but there is a lack of content and further analysis related to neutral lipids and polar lipids. The amount of information available for readers' reference is too small to meet the publishing requirements.
Thank you for your consideration. However, at lines 114-117 of the introduction, authors said: “The principal aim of this study was to evaluate the total lipid content and to analyze the fatty acid profile of liver and ovary tissues from immature and mature aquaculture Moroccan fishes Hypophthalmichthys molitrix and Cyprinus carpio, which were grown in the same conditions and compare their potential nutritional value with the fatty acid profile of their edible fillets”.
The core content of this work was the fatty acids profiles of different fish tissues. Authors extracted from gonads, liver and fillet the total lipids, as reported in table 1 but, as described in Materials and Methods, the total lipid extracts were saponified and derivatized and after a proper reaction and extraction , as reported in materials and methods, we recovered and analyzed the fatty acids methyl esters profiles. The attention was focused on this particular class of lipids; any chromatographic separation procedure of single lipid classes was not performed because it was not the focus of the work.
In any case the text has been modified for a better understanding of the focus of this manuscript.
- The class and contents of main fatty acids should be listed in Table 1, rather than detailed in the text of the results.
In table 1 the authors reported in the columns only the total amount of the SFA, MUFA and for the PUFA in particular the w-3 and w-6 values. The text has been simplified and authors have added in supplementary materials a further table where values of the most important fatty acids are collected for each class.
- AI and TI are commonly used indicators in animal experiments, and their use in this study is inappropriate.
Thank you for your precious considerations, but we would like to say that the AI and TI indices are also used to determine the nutritional quality of fish tissues; in fact, in the text the authors have cited some articles (e.g. bibliography 50, 53,59) and others can be found in the literature. Authors suggest that this index could be very useful into the nutritional characterization of investigated edible fish tissues (fillet) as well as waste fish tissues (ovary and liver).
- For statistical methods, “student-t” cannot be used for statistical processing between more than three groups of experimental data.
Thank you for your comment. We understand that we have more than two groups of data (two fish species, each at two growth stage) and this could be misleading. Actually, we were not interested in comparing multiple classes: for each lipid species, we only assessed the differences between common and silver carp within the same growth stage, i.e. mature or immature. For this reason, we used the Student's t-test and not the ANOVA test with post-hoc adjustments.
This aspect has been better specified in the text.

This manuscript is a resubmission of an earlier submission. The following is a list of the peer review reports and author responses from that submission.
Round 1
Reviewer 1 Report
The manuscript showed results of fatty acids from parts of fishes but no information about diet was supplied to support the conclusion. The authors failed to show statistical comparison among results. The analysis of fatty acids was performed by CG/MS with a column not suitable and the conditions of quantification were not reported. The formula used should be revised.
Summary - What does marine environments mean in the summary? Please see lines 112-114 page “Specimens of both species were recovered from Deroua fisheries farm at Beni Mellal-Khenifra region, which is one of the few fish farming hubs not along Morocco’s coast and where there is a flourishing aquaculture sector”
Is Ovary an edible part?
Introduction – lines 88 – 92 – Please review “marine fishes are a precious source of health-beneficial PUFA, mainly eicosapentaenoic acid (EPA), docosahexaenoic acid (DHA)” and “the essential fatty acids are linoleic and linolenic acids”.
Results and discussion:
According to Section 4.1 “Hypophthalmichthys molitrix (silver carp) and Cyprinus carpio (common carp) were recovered from Deroua Fisheries Station located in Beni Mellal-Khenifra region, a semi-arid climate zone in Morocco. The collection was done in April 2019. Fishes were held both in earthen ponds where silver carp are fed with natural phytoplankton and common carp is omnivorous.
The results of lipids and fatty acids depended on the fish diet and no information about that was supplied to support the conclusion.
The analysis of fatty acids were performed by GC/MS using a column OV-5 column, VF-5ms 30 x 0.25, Agilent which is not currently used for FAME analysis because there are more suitable polar columns and the GC/MS is not suitable quantification. No information was supplied about CG/MS if scan mode or selective ion monitoring (SIM), with calibration curves built for a single ion from each compound to be quantified.
The authors failed to show statistical comparison among results.
Please explain – significant discrepancies (discrepancy|/error>1 in the table 1)
It is not possible to calculate the mg of fatty acids/100 grams of fillet using the formula present in the line 505 because the weight of lipids/100g fillet contain fatty acids esterified to glycerol. Please see “the it was possible to establish the total amount of each fatty acid (FA) contained in the recovered EE extract of each fish species; according to the weight of the frozen fillet fish tissue (mfis), the mgs of each fatty acid in 100 g of fish tissue were determined by the following formula:
[(mg FA in 1mg of EE extract x total mg of EE extract)/g mfis] x 100
Reviewer 2 Report
The manuscript compared the content of fatty acids in the Moroccan Hypophthalmichthys molitrix and Cyprinus carpio. The manuscript lacks of novelty except for the identification of the fatty acids in fish. This is very basic research and may not contribute to the field. Even though there is a point that may contribute to readers, which is the various fatty acids content in immature and mature female carp. However, the various fatty acids content in the gonads, liver and fillet tissues make me confused. In general, consumers mainly consume fish fillets. Gonads and liver are not included in the scope of consumption. In addition, gonads and livers are mixed with other internal organs of the fish in small proportions and are difficult to take out. They have little medicinal value. Why not consider fish bladder and gills?
Overall, I recommended a rejection of the manuscript mainly due to the lack of novelty and data, and the information would not give other researchers effective inspiration.
Line 22-23, Will Moroccan Hypophthalmichthys molitrix and Cyprinus carpio fishes survive in seawater? As far as I know they are supposed to be freshwater fish.
Reviewer 3 Report
Interesting MS and interesting research on topics important to consumers. Aquatic foods are often referred to as "functional foods". This MS is part of this research trend. However, it needs improvement before final approval. Due to the large amount of details that need improvement, I suggest a major revision. However, I believe that the authors will improve this MS without problems. My detailed comments are included in the text. To see them all, open the file in Acrobat Reader.
Please remember that each paragraph should consist of at least three sentences. Please combine these mini-paragraphs (especially in Discussions) into larger units for each discussed aspect of MS.

Reviewer 4 Report
Development of accurate nutritional information of aquaculture products provides the data base for nutritional labelling, better processing and marketing.
Discussion on mobilization of fatty acids to gonad has to be improved. Fish fatty acid profiles have to be correlated with the feed for better understanding of the retention, preferential metabolism of individual fatty acids.
would have Why completeThe data presented in this MS is pro